# A First-Principle Study of Interactions between Magnesium and Metal-Atom-Doped Graphene

**DOI:** 10.3390/nano12050834

**Published:** 2022-03-01

**Authors:** Yaoming Li, Xin Pei, Huang Zhang, Meini Yuan

**Affiliations:** 1School of Mechanical Engineering, North University of China, Taiyuan 030051, China; lym@nuc.edu.cn (Y.L.); zhanghuang@nuc.edu.cn (H.Z.); 2The College of Mechatronic Engineering, North University of China, Taiyuan 030051, China; px2565586030@163.com

**Keywords:** density functional theory, doped graphene, magnesium, interactions, interface, graphene/Mg(001)

## Abstract

In this study, the interactions of magnesium (Mg) atom and Mg(001) surface with different metal-atom-doped graphene were investigated using a density functional theory (DFT) method. For the interactions of magnesium with Al-, Mn-, Zn-, and Zr-doped and intrinsic graphene, it was found that the magnesium atoms were physisorbed into the hollow sites of the intrinsic graphene with only the smallest interaction energy (approximately −1.900 eV). However, the magnesium atoms tended to be chemisorbed on the doped graphene, which exhibited larger interaction energies and charge transfers. Additionally, the Zn-doped graphene displayed the largest interaction energy with the Mg atom (approximately −3.833 eV). For the interactions of Mg(001) with Al-, Mn-, Zn-, and Zr-doped and intrinsic graphene (intrinsic and doped graphene/Mg interface), doped atoms interacted with a Mg layer to make graphene wrinkle, resulting in a higher specific surface area and better stability. Mg–C chemical bonds were formed at the Al-, Zn-, and Zr-doped interface, and Mg–Mn chemical bonds were formed at the Mn-doped interface. This study provided the fundamental research for future research into doped atoms on graphene reinforced magnesium matrix composites.

## 1. Introduction

Graphene is currently one of the most prominent nanoscale materials due to its excellent mechanical, structural, and electrical properties [1]. Doping some atoms in graphene can well regulate the microscopic properties of graphene. For example, Pooja Rani et al. [2,3,4,5] systematically studied the doping of B atom and N atom in graphene to adjust the band gap using DFT methods. Additionally, the interactions of graphene and metal-atom-doped graphene have become hot issues in the material field. For example, Caragiu et al. [6,7,8,9] studied the interactions between alkali metals and graphene. Additionally, transition and noble metals that were adsorbed on graphene sheets were previously examined [10,11,12,13,14,15]. It is believed that magnetic metals that are doped on graphene [16,17,18,19,20,21] have the abilities to induce large magnetic energy, or strong covalent bond formations, which can modify the electronic and magnetic properties of intrinsic graphene.

Graphene-reinforced magnesium matrix composites have wide applications in both aerospace and aeronautical fields [22]. Therefore, it is necessary to reveal the interaction between graphene system and magnesium atoms. In recent years, some researchers have studied the interaction between graphene system and magnesium species using computational simulation methods. For example, Kato et al. [23,24] used a DFT method to investigate the interactions of the magnesium species (Mg, Mg^+^, and Mg^2+^) on graphene. The results showed that the binding nature of the Mg ions (Mg^+^ and Mg^2+^) was caused by charge transfer interactions. Additionally, the Mg atoms were observed to interact with the graphene surfaces via a van der Waals (vdW) force. The distances between the Mg atoms and the graphene surfaces were calculated to be 1.80 Å (Mg^2+^), 2.16 Å (Mg^+^), and 4.17 Å (Mg). In a related study, Tachikawa et al. [25] used direct molecular orbital molecular dynamics (MO–MD) to calculate the interactions of magnesium atoms and graphene surfaces. The results of their study showed that the Mg atoms were bound to the hollow site located at 2.02 Å from the graphene surface. The Mg atoms were observed to vibrate within the hollow site, and diffusion did not occur, even at 1000 K. However, it is noted that there is little research on the interaction mechanism between metal-atom-doped graphene and magnesium from the perspective of theoretical calculation.

In this study, a DFT method was used to calculate the interaction energy, total electron density, and electron density differences between the magnesium atoms and the graphene or doped graphene (Al-, Mn-, Zn-, and Zr-doped). Based on the interactions of magnesium atoms with intrinsic and doped graphene, the interface properties of intrinsic graphene/Mg(001) and doped graphene/Mg(001) are revealed by calculating the interface adhesion work, electron density, Mulliken populations, and partial density of states (PDOS) using a DFT method. This study could reveal the interaction mechanism between magnesium and doped graphene through DFT calculation, and provide the fundamental research for future research into doped atoms on graphene-reinforced magnesium matrix composites.

## 2. Computational Details

In this research study, all of the calculations were performed using an efficient ab initio computer code: DMol3. This code was used to investigate the interactions between the magnesium atoms and the doped or intrinsic graphene. A double-numerical plus polarization (DNP) basis set was employed to produce highly accurate results, while keeping the costs of the computational processes fairly low. A DFT semicore pseudopotential (DSPP) was used to describe the core electrons [26]. A generalized gradient approximation (GGA) was used for exchanging the functional correlation, as described by Perdew–Burke–Ernzerhof (PBE). A Fermi smearing of 0.005 Ha (1 Ha = 27.2114 eV) was used. The convergence criteria for the geometric optimization and energy calculation were set as follows: a self-consistent field tolerance of 1.0 × 10^−6^ Ha/atom, energy tolerance of 1.0 × 10^−5^ Ha/atom, maximum force tolerance of 0.002 Ha/Å, and maximum displacement tolerance of 0.005 Å [27]. For the accurate description of the processes studied, it is necessary to consider the van der Waals forces. Therefore, the Grimme method for DFT-D correction was used.

In a DMol3 module, the parameters to be tested for convergence include lattice constant, k-point, and orbital cutoff. As can be seen from Figure 1, the optimal lattice constant (a) is 2.4675 Å, which is consistent with the experimental value (2.46 Å) [2]. Monkhorst–Pack schemes with 6 × 6 × 1 k-point mesh were used for the special point sampling in the Brillouin zone. A global orbital cutoff of 6.0 Å was employed. As shown in Figure 1d, for pure graphene, the calculated band structure clearly shows that at the H point and K point, the conduction band and valence band intersect at the Fermi level (energy is zero). Therefore, pure graphene is a semiconductor with a zero band gap.

## 3. Results and Discussion

### 3.1. Interaction of Magnesium Atom with Intrinsic and Doped Graphene

#### 3.1.1. The Adsorbed Structural Properties

The system was modeled as a 4 × 4 graphene supercell, which contained 32 atoms with a periodic boundary condition. A vacuum space of 60 Å was set in the normal direction to the sheets in order to avoid interactions between the periodic images. The optimized structural model of the intrinsic graphene is shown in Figure 2a. A model of the doped graphene was built based on the model of the intrinsic graphene. For the choice of doped metal atoms, Al, Mg, Zn, and Zr are considered doped atoms because the common alloy elements (Al, Mg, Zn, and Zr) in magnesium alloys [28] are well combined with magnesium. One C atom was replaced with an Al, Mn, Zn, or Zr atom in the doped graphene model. Then, geometry optimization was carried out on both of the models, as shown in Figure 2b–e.

The energy variations in the doped graphene were systematically studied in order to investigate the effects of the doped atoms (Al, Mn, Zn, and Zr) on the microstructures and electronic structures of the graphene. The energies of a single doped atom and C atom were calculated. It is clear that the energy of the doped atom (Al, −6591.71 eV; Mn, −3312.10 eV; Zn, −6922.08 eV; Zr, −1939.40 eV) is greater than that of the substituted C atom (−1028.68 eV). As shown in Table 1, it was observed that the doped atoms increased the energy of the graphene by approximately 2% to 16%. From the energy point of view, the Zn atom contributes the most to the energy of graphene. It can also be seen in Table 2 that the initial bond length of the C–C was approximately 1.425 Å, which was found to be close to the previous results [29]. The C–C bonds around the defects in the doped structures were observed to be either compressed or stretched. By taking the Zn-doped graphene as an example for the analysis, it was determined that when the C atom was replaced by the Zn atom, the bond length of the Zn–C increased to approximately 1.722 Å. The results were determined to be similar to those obtained in a previous study performed by Zhang et al. [29]. Then, in order to determine any further effects of the doped atoms on the graphene, the charge changes were also explored. It was found that, after being replaced by an Al, Mn, Zr, or Zn atom, the C atom that bonded to the doped atoms became negatively charged.

In this research study, a Mg atom was added above the intrinsic and doped graphene in order to investigate the influences of the different doped graphene on the Mg atoms. Then, in order to obtain the most stable interaction configuration, the Mg atom was initially placed at different positions above the graphene. Following a relaxation period, the optimized configurations obtained from the different initial states were compared for the purpose of identifying the most favorable state.

The configurations of the modeling systems following the DFT calculation are shown in Figure 3. It can be seen from the figure that the geometric structure of the doped graphene was dramatically changed. Then, by taking the Al-doped graphene as an example, the bonds (B1, B2, and B3) around the Al atom were found to be elongated from 1.722, 1.722, and 1.722 Å to 1.974, 1.974, and 1.974 Å, respectively, as shown in Table 2. As can be seen in Table 3, the bond lengths of the Al–Mg, Mn–Mg, Zn–Mg, Zr–Mg, and C–Mg in three directions following the Mg atom interactions were calculated in order to illustrate the configurations of the interaction system.

The bond length of the C–Mg between the Mg and the intrinsic graphene was determined to be 3.029 Å. This was too long to form any chemical bonds. Additionally, the structures experienced small charge transfer values, which confirmed that there was only a weak interaction between the intrinsic graphene and the Mg atoms (Table 4). As previously mentioned, it was observed that the interactions were weak physical interactions due to a van der Waals interaction between the Mg atoms and the intrinsic graphene. These results were found to agree well with the results previously reported by Tachikawa et al. [25].

#### 3.1.2. Interaction Energy of Magnesium with Doped and Intrinsic Graphene

The Eads indicated the interaction intensity between the Mg atoms and the intrinsic or doped graphene, and was derived according to the following equation:E_ads_ = E_graphene+Mg_ − (E_graphene_ + E_Mg_),(1)
where E_graphene+Mg_, E_Mg_, and E_graphene_ represent the system total energy, the Mg atoms energy, and the graphene sheet energy, respectively. A negative Eads value corresponded to the stable interactions. The higher the negative Eads value was, the more stable the adsorbed structure was.

The E_ads_ between the Mg atoms and the intrinsic graphene was approximately −1.900 eV (Table 4). The doped atoms (Al, Mn, Zn, and Zr) were found to greatly increase the Eads. In this study, by taking the Zn-doped graphene as an example, it was determined that the E_ads_ between the Zn-doped graphene sheet and Mg atoms was approximately −3.833 eV. Additionally, the bond length of the C–Mg between the Zn-doped graphene and the Mg atoms was observed to have majorly decreased (2.161 Å), as shown in Table 3. With the greatly increasing interaction energy, the interactions of the Mg with the Zn-doped grapheme became much stronger. The relatively large binding energies, along with the small binding distance of these doped graphene adsorption systems, indicated that chemical bonds were formed between Mg and doped graphene. Therefore, the interactions between the Mg atoms and the doped graphene could potentially become majorly strengthened. Among them, the longest C–Mg bond in Mn-doped graphene (4.028 Å) is not enough to form a chemical bond, while the shortest C–Mg bond in Zn-doped graphene (2.161 Å) forms a stable chemical bond. Additionally, the interaction energy of Zn-doped graphene (−3.833 eV) is higher than that of Mn-doped graphene (−2.590 eV). Therefore, the Zn-doped graphene had the largest capability to capture the Mg atom, whereas the Mn-doped graphene displayed the lowest capability to capture the Mg atom. It was also concluded that the interactions between the doped graphene and the magnesium matrix composites could be greatly increased in strength.

#### 3.1.3. Electronic Properties of Magnesium with Doped and Intrinsic Graphene

The electron densities, as well as the electron density differences, were examined in this study for the purpose of illustrating the electron transfers during the interactions of the Mg atoms and the intrinsic or metal-atom-doped graphene. As shown in Figure 4, the isosurface (isovalue: +0.15 e/Å^3^) of the total electron density indicated that the Zn-doped graphene and Mg atoms had a high strength joint, as well as the largest isosurface area. This was followed by the Al-, Zr-, and Mn-doped graphene sheets. These results were found to be in accordance with the Eads values that had been previously calculated (3.1.2: Interaction energy of magnesium with doped and intrinsic graphene). Although an overlapping of the electrons was observed between the Mg atoms and the doped graphene sheets, detailed electron transfers were not observed. Therefore, the electron density differences between intrinsic and atom-doped graphene were used to illustrate the gain and loss of electrons during the interactions. As shown in Figure 5e,f (isovalues: yellow = −0.02 e/Å^3^; blue = +0.02 e/Å^3^), the electron densities of both the Mg atoms and the Zn atoms demonstrated electron losses. These lost electrons formed intense interactions between the doped atom, Mg atom, and C atoms in the Zn-doped graphene. The electron accumulation areas were observed to be small in the Al-, Zr-, and Mn-doped graphene, as detailed in Figure 5. The losses and gains of the electrons could also be determined by the Mulliken charge calculations before and after the Mg atom interactions, as listed in Table 5. The charge transfer from the Mg atoms before and after adsorption on the Zn-doped graphene was the largest (0.806 e), which proves that the interaction between Zn-doped graphene and magnesium atoms was the strongest and consistent with the maximum interaction energy (−3.833 eV).

### 3.2. Interaction of Mg(001) Surface with Intrinsic and Doped Graphene

#### 3.2.1. Interface Structures

The graphene/Mg interface microstructure has a significant impact on the properties of the composites. Therefore, the interface properties of the intrinsic graphene/Mg and four-doped graphene/Mg were calculated based on the interactions of magnesium atom with intrinsic and doped graphene. In the study of magnesium, the Mg(001) surface can be used as the preferred surface of interface structure [30,31]. As shown in Figure 6, the intrinsic interface model includes five layers of atoms, one layer of graphene (4 × 4) supercell (32 atoms), and four layers of Mg(001)−(3 × 3) supercell (36 atoms). By doping Al, Mn, Zn, and Zr atom in the graphene layer of the intrinsic interface model, the doped graphene/Mg(001) interface models were established. Additionally, the lattice mismatch ratios of the five interface models are less than 1%. In the calculation, the lowest two layers of atoms were fixed, the other atoms were completely relaxed, and the initial interface spacing was set to 2 Å. In addition, the calculation parameters of the interface structure are the same as those in the calculation of adsorption.

Figure 7 shows the top and side views of the five interface models after geometry optimization. Compared with the graphene layer flatness at the intrinsic graphene/Mg interface, the graphene layers at the doped interfaces become uneven, and the corresponding Mg atoms are slightly closer to the graphene layer, which causes the chemical bonds in the interface model to be stretched or compressed. Compared with magnesium interactions of the intrinsic and doped graphene sheets, the moving directions of doped atoms are consistent; for example, an Al atom, Zn atom, and Zr atom are far away from a Mg atom, while a Mn atom is near a Mg atom. Unlike the adsorption model, C atoms of a graphene layer in the doped interface models are also close to a Mg atom due to the interaction of the doped atoms.

#### 3.2.2. Interface Adhesion Work of Mg(001) with Doped and Intrinsic Graphene

Interface adhesion work is an important index to measure the strength of interface stability, which is calculated by the following formula [32]:Wad = (E_graphene_ + E_Mg(001)_ − E_interface_)/A(2)
Here, E_graphene_, E_Mg(001)_, and E_interface_ represent the energy of graphene, four-layer Mg(001) slab model, and interface model, respectively, and A represents the interface area. The larger the interface adhesion work is, the smaller the interface spacing is, which means the stronger the interface stability is. As shown in Table 6, compared with the intrinsic graphene/Mg interface model, the interface stability of the doped interface model is higher than that of the intrinsic graphene/Mg interface model, and the interface spacing is smaller than that of the intrinsic graphene/Mg interface model. Therefore, it can be seen that the doped atoms make the interface stability stronger. The Zr-doped interface model has the largest interface adhesion work, then the Mn-, Zn-, and Al-doped interface, which is different from the interaction energy between the Mg atom and the graphene sheets. No chemical bonds are formed at the intrinsic graphene/Mg interface model because of the maximum interface spacing (3.8801 Å). Combined with Figure 7, it can be clearly seen that the doped atoms lead to the surface puckering of graphene, which increases the specific surface area of graphene and also forms chemical bonds at the interface, which gives a higher work of adhesion at the doped interface.

#### 3.2.3. Electronic Structure of Mg(001) with Doped and Intrinsic Graphene

To understand the electronic interactions, charge transfer, and bond forming characteristics at the interfaces, the electron density, Mulliken populations, and partial density of states of the five interface models were investigated, respectively.

Figure 8 shows isosurfaces of the electron density of five interface models. When the isovalue is +0.15 e/Å^3^, compared with the intrinsic graphene/Mg interface, the electron densities of the Mg atoms in the Mg-1 layer in the doped interfaces overlap with the doped graphene, which indicates that there is chemical bond formation at the interface. In order to quantitatively analyze the charge transfer at the interface, Mulliken populations of five interface models are analyzed, as shown in Table 7. In general, the Mg-1 layer gets the charge, and the Mg-2 layer, C1, C2, and C3 lose the charge. In the doped interface models, the Al atom, Zn atom, and Zr atom getting charge tend to be far away from the Mg atom, the corresponding C atom with a charge loss of more than 0.4 tends to combine with the Mg atom to form the Mg–C chemical bond, while the Mn atom with a charge loss tends to be close to the Mg atom to form the Mg–Mn chemical bond.

In order to further understand the bonding characteristics at the interface, partial densities of the state of the five interface models were calculated and shown in Figure 9. On the whole, the electronic states at the Fermi level (E_f_) of the graphene layer in the intrinsic graphene/Mg interface model are zero, indicating that the existence of a magnesium matrix does not make the graphene layer change into metallicity, while the electronic states at the Fermi level of the graphene layer in the doped interface are not zero (very small), indicating that the doped atoms can change the graphene layer into metallicity. In the Al-doped interface (Figure 9a), the C atoms and Al atom of a graphene layer have the same peak tendency near the energy (−18.6 eV), indicating that the Al atom is suitable for doping in graphene, and the C atoms and Mg atoms in the Mg-1 layer have the same sharp peaks near the energy (−4.7, −4.0, −0.5 eV), showing that Mg–C chemical bonds are generated. In the Mn-doped interface (Figure 9b), the d orbital of the Mn atom and the p orbital of the Mg atom in the Mg-1 layer have the same peak tendency near the Fermi level and energy (−6.2, −4.0, and −2.7 eV), indicating that Mg–Mn chemical bond is generated and the interface is relatively stable. In the Zn-doped interface (Figure 9c), the C atoms and Mg atoms of the Mg-1 layer exhibit the same peak tendency near the energy (−18.7, −6.6, and −2.2 eV), implying that Mg–C chemical bonds are formed. In the Zr-doped interface (Figure 9d), the C atoms and Zr atom of a graphene layer have the same peak tendency in many ranges, indicating that the Zr atom is also suitable for doping in graphene, and the C atoms and Mg atoms in the Mg-1 layer have the same sharp peak tendency in many ranges, implying that Mg–C chemical bonds are generated. In the intrinsic graphene/Mg interface (Figure 9e), the densities of state peaks of the C atom and the Mg atom have almost no similar part, showing that no chemical bonds are formed at the intrinsic graphene/Mg interface model.

The chemical bonds are formed at the doped interface, but no chemical bonds are formed at the intrinsic graphene/Mg interface. In the Al-, Zn-, and Zr-doped interface models, a Mg–C chemical bond is formed at the interface, while in the Mn-doped interface model, a Mg–Mn chemical bond is formed at the interface. Additionally, the formation of chemical bonds can improve the stability of the interface.

## 4. Conclusions

The interaction abilities of an isolated Mg atom and Mg(001) surface with the intrinsic and doped graphene were theoretically investigated using DFT methods. The interactions between the Mg atom and the Mg(001) surface on intrinsic and doped graphene are revealed by calculating the energy, electron density, electron density differences, and partial density of state of the model system. The specific results are as follows:For the interactions of a magnesium atom with Al-, Mn-, Zn-, and Zr-doped and intrinsic graphene, the bond lengths of the doped metallic atoms with the surrounding C atoms were stretched or compressed, which greatly changed the local structure of graphene. The interaction energy of Zn-doped graphene (−3.833 eV) was higher than that of Mn-doped graphene (−2.590 eV). The result indicated that the Zn-doped graphene had the largest capability to capture the Mg atom, whereas the Mn-doped graphene displayed the lowest capability to capture the Mg atom. Additionally, the electron transfers between the doped graphene and the Mg atom were examined. The results indicated that the interaction between Zn-doped graphene and magnesium atoms was strongest.For the interactions of a Mg(001) surface with Al-, Mn-, Zn-, and Zr-doped and intrinsic graphene (intrinsic and doped graphene/Mg interface), the doped atoms are far away from or near the Mg atoms because of the influence of several Mg atoms on the Mg(001) surface, and the constraint of the doped atom–C chemical bond results in a great change in the local structure around the doped atoms, resulting in the wrinkles of the graphene layer, which also increases the specific surface area of the graphene layer. The results indicated that the doped atoms could improve the interface adhesion work and thus make the interface more stable. There was no chemical bond at the intrinsic graphene/Mg interface due to the maximum interface spacing (3.8801 Å). Mg–C chemical bonds were formed at the Al-, Zn-, and Zr-doped interface, and a Mg–Mn chemical bond was formed at the Mn-doped interface.Compared with the interactions of a magnesium atom with intrinsic and doped graphene, the movement directions of doped atoms are consistent in the interface model. For example, an Al atom, Zn atom, and Zr atom are far away from Mg atoms due to the gain of charge, and Mn is close to Mg atoms due to the loss of charge.

## Figures and Tables

**Figure 1 nanomaterials-12-00834-f001:**
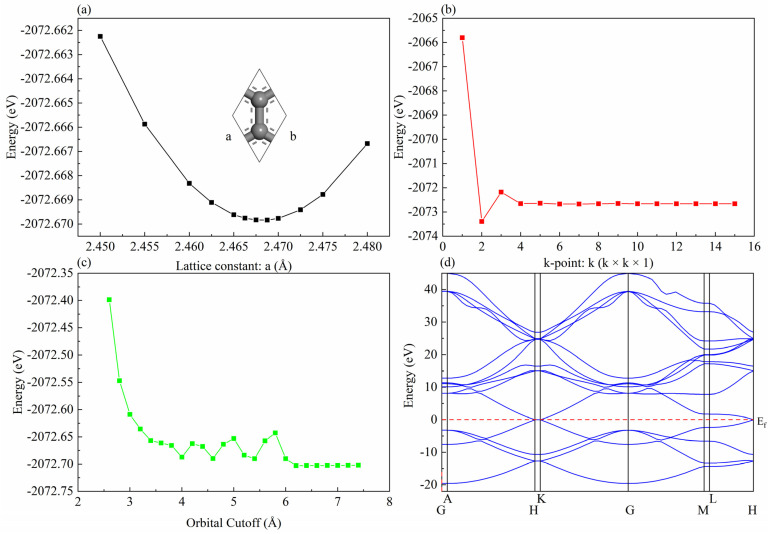
The convergence tests: (**a**) lattice constant, (**b**) k-point, (**c**) orbital cutoff, (**d**) band structure.

**Figure 2 nanomaterials-12-00834-f002:**
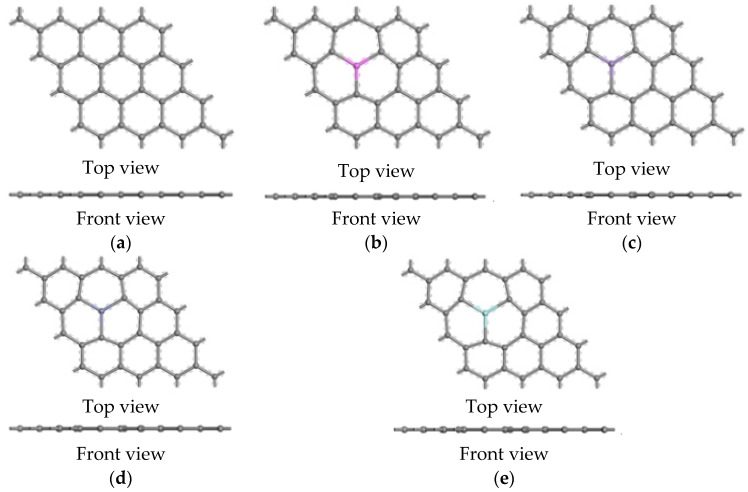
Ball and stick model of the intrinsic and doped graphene following the DFT calculation. (**a**) Intrinsic graphene sheet, (**b**) Al-doped graphene sheet, (**c**) Mn-doped graphene sheet, (**d**) Zn-doped graphene sheet, (**e**) Zr-doped graphene sheet.

**Figure 3 nanomaterials-12-00834-f003:**
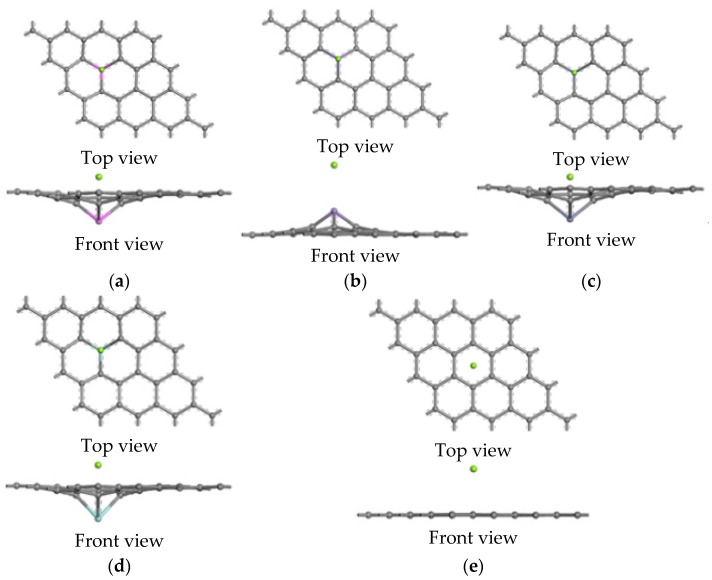
The interactions of magnesium with the intrinsic and doped graphene sheets. (**a**) Al-doped graphene sheet, (**b**) Mn-doped graphene sheet, (**c**) Zn-doped graphene sheet, (**d**) Zr-doped graphene sheet, (**e**) Intrinsic graphene sheet.

**Figure 4 nanomaterials-12-00834-f004:**
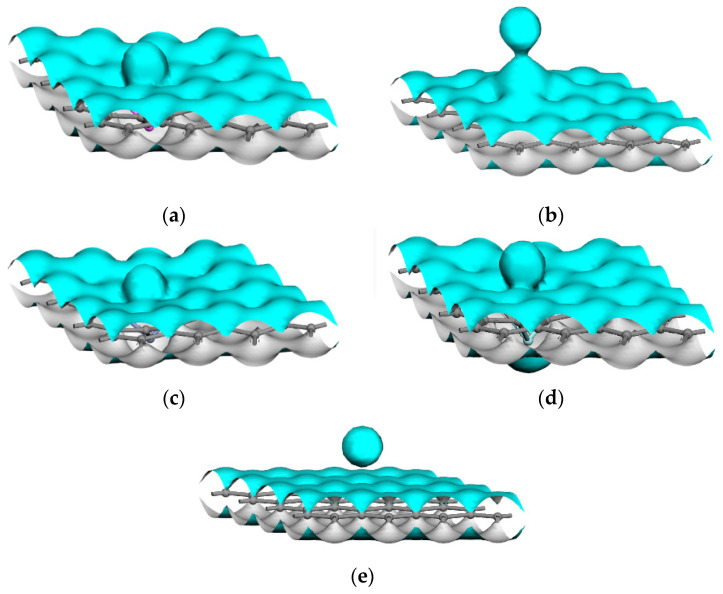
Isosurfaces of the electron densities of the Mg interactions with the intrinsic and doped graphene sheets: (**a**) Al-doped graphene sheet, (**b**) Mn-doped graphene sheet, (**c**) Zn-doped graphene sheet, (**d**) Zr-doped graphene sheet, (**e**) intrinsic graphene sheet.

**Figure 5 nanomaterials-12-00834-f005:**
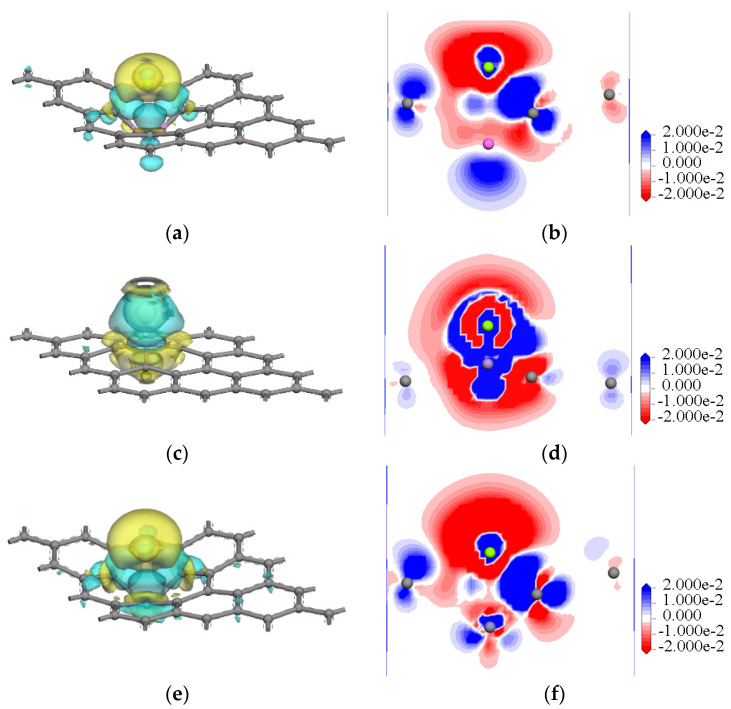
Three-dimensional forms and slices of the electron density differences for the Mg interactions with the intrinsic and doped graphene sheets: (**a**) 3D: Al-doped graphene sheet; (**b**) slice: Al-doped graphene sheet; (**c**) 3D: Mn-doped graphene sheet; (**d**) slice: Mn-doped graphene sheet; (**e**) 3D: Zn-doped graphene sheet; (**f**) slice: Zn-doped graphene sheet; (**g**) 3D: Zr-doped graphene sheet; (**h**) slice: Zr-doped graphene sheet; (**i**) 3D: intrinsic graphene sheet; (**j**) slice: intrinsic graphene sheet.

**Figure 6 nanomaterials-12-00834-f006:**
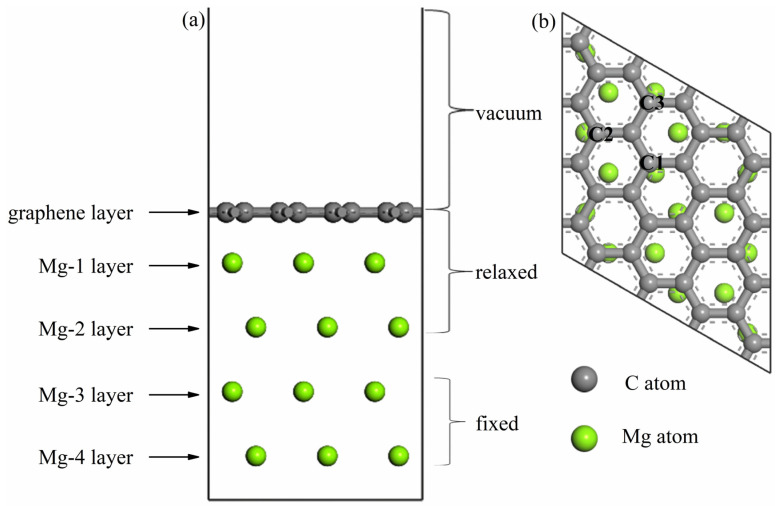
The intrinsic interface model: (**a**) side view; (**b**) top view (the position of the C atom surrounded by C1, C2, and C3 is the position of the doped atom).

**Figure 7 nanomaterials-12-00834-f007:**
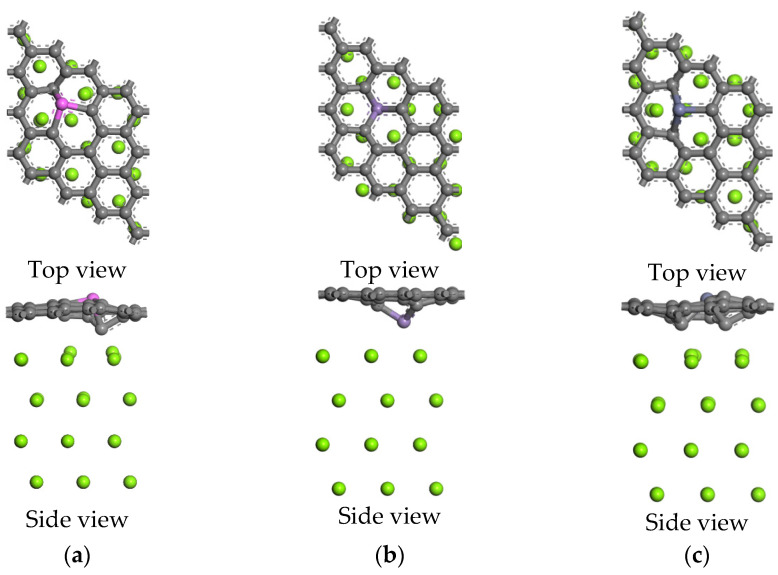
The top and side views of the five interface models after geometry optimization: (**a**) Al-doped interface, (**b**) Mn-doped interface, (**c**) Zn-doped interface, (**d**) Zr-doped interface, (**e**) intrinsic interface.

**Figure 8 nanomaterials-12-00834-f008:**
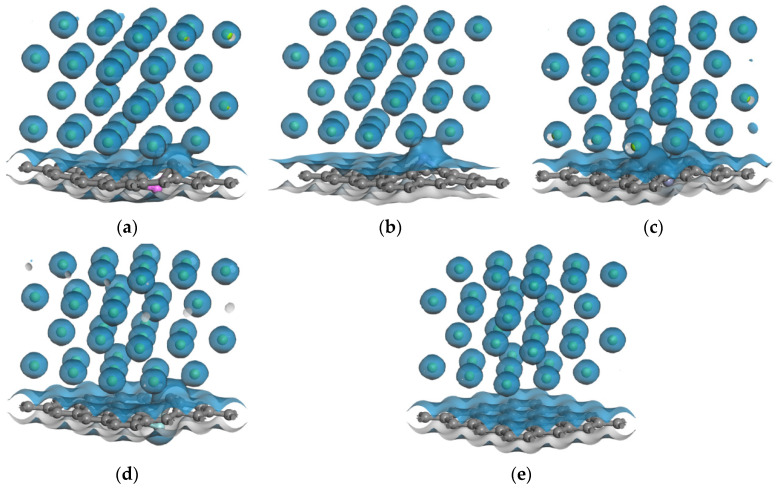
Isosurfaces of the electron densities of five interface models: (**a**) Al-doped interface, (**b**) Mn-doped interface, (**c**) Zn-doped interface, (**d**) Zr-doped interface, (**e**) intrinsic interface.

**Figure 9 nanomaterials-12-00834-f009:**
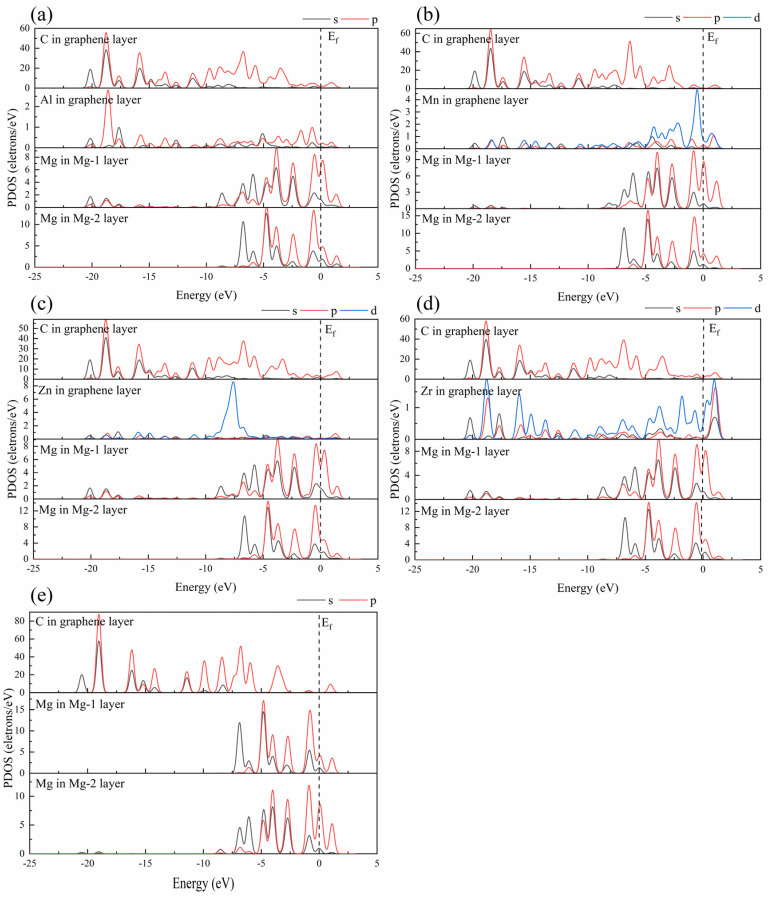
The partial density of the state of five interface models: (**a**) Al-doped interface, (**b**) Mn-doped interface, (**c**) Zn-doped interface, (**d**) Zr-doped interface, (**e**) intrinsic interface.

**Table 1 nanomaterials-12-00834-t001:** Energies of the intrinsic graphene and doped graphene (Al, Mn, Zn, and Zr) following the DFT calculation (unit: eV).

Graphene	Intrinsic	Al-Doped	Mn-Doped	Zn-Doped	Zr-Doped
Energy	−33,162.54	−38,713.16	−35,433.05	−39,039.06	−34,056.45
Deviation	-	16.74%	6.85%	17.72%	2.70%

**Table 2 nanomaterials-12-00834-t002:** Bond lengths of the C–C, Al–C, Mn–C, Zn–C, and Zr–C in three directions (B1, B2, and B3) before and after the interactions (unit: Å).

Bond Length	B1		B2		B3	
	Before Interaction	After Interaction	Before Interaction	After Interaction	Before Interaction	After Interaction
Intrinsic	1.425	1.423	1.425	1.427	1.425	1.427
Al-doped	1.707	1.920	1.707	1.920	1.707	1.920
Mn-doped	1.671	1.790	1.671	1.790	1.671	1.790
Zn-doped	1.722	1.974	1.722	1.974	1.722	1.974
Zr-doped	1.845	2.088	1.845	2.088	1.845	2.088

**Table 3 nanomaterials-12-00834-t003:** Bond lengths of the Al–Mg, Mn–Mg, Zn–Mg, Zr–Mg, and C–Mg in three directions following the Mg interactions (unit: Å).

Bond Length	Doped Atom-Mg	C1-Mg	C2-Mg	C3-Mg
Intrinsic	3.032	-	3.029	3.031
Al-doped	2.716	2.268	2.268	2.268
Mn-doped	2.788	4.028	4.028	4.028
Zn-doped	2.564	2.161	2.161	2.161
Zr-doped	3.204	2.376	2.376	2.376

**Table 4 nanomaterials-12-00834-t004:** Interaction energy between the Mg and the graphene sheets (unit: eV).

Graphene	Intrinsic	Al-Doped	Mn-Doped	Zn-Doped	Zr-Doped
Interaction energy	−1.900	−3.715	−2.590	−3.833	−2.858

**Table 5 nanomaterials-12-00834-t005:** Mulliken atomic charges (MACs) of the atoms in the intrinsic and doped graphene.

Graphene System		Mg	Doped Atom	C1	C2	C3
MAC of intrinsic graphene	Before interaction	0.000	0.000	0.000	0.000	0.000
	After interaction	0.125	−0.032	0.014	−0.030	−0.031
MAC of Al-doped graphene	Before interaction	0.000	0.574	−0.340	−0.328	−0.362
	After interaction	0.619	0.575	−0.495	−0.496	−0.486
MAC of Mn-doped graphene	Before interaction	0.000	−0.279	−0.037	−0.029	−0.059
	After interaction	0.232	−0.434	−0.016	−0.012	−0.025
MAC of Zn-doped graphene	Before interaction	0.000	−0.238	−0.115	−0.071	0.059
	After interaction	0.806	0.200	−0.386	−0.388	−0.371
MAC of Zr-doped graphene	Before interaction	0.000	0.758	−0.377	−0.381	−0.355
	After interaction	0.479	0.575	−0.486	−0.484	−0.471

**Table 6 nanomaterials-12-00834-t006:** Interface adhesion works (W_ad_) and interface spacings (d_0_) of five interface models.

Interface System	Intrinsic	Al-Doped	Mn-Doped	Zn-Doped	Zr-Doped
W_ad_ (J/m^2^)	0.0631	0.1163	0.7537	0.2823	1.2801
d_0_ (Å)	3.8801	2.8604	3.2576	2.9036	3.1092

**Table 7 nanomaterials-12-00834-t007:** Mulliken atomic charges (MAC) of the atoms in five interface models.

Interface System	Mg-1 Layer	Mg-2 Layer	Doped Atom	C1	C2	C3
Intrinsic	0.052–0.053	−0.026	−0.007	−0.007	−0.007	−0.007
Al-doped	0.074–0.244	−0.015–0.012	0.607	−0.282	−0.282	−0.662
Mn-doped	0.070–0.120	−0.021–−0.016	−0.696	−0.064	−0.071	−0.065
Zn-doped	0.055–0.226	−0.019–0.005	0.145	0.145	−0.483	−0.493
Zr-doped	0.063–0.231	−0.017–0.008	0.622	0.622	−0.226	−0.481

## Data Availability

The original contributions presented in the study are included in the article. Further inquiries can be directed to the corresponding author.

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
