# Peer review of "A First-Principle Study of Interactions between Magnesium and Metal-Atom-Doped Graphene"

_nanomaterials, 2022, doi:10.3390/nano12050834_

Round 1
Reviewer 1 Report
The responses to the questions seem to me to be quite detailed and satisfactory. So, in my opinion, the authors responded to all comments and recommendations, and the manuscript can be accepted for publication.
Author Response
Thank you very much for your valuable suggestions. After checking grammar and spelling, all such expressions have been corrected:
- We revised “a global orbital cutoff of 6.0 Å were emplloyed” as “A global orbital cutoff of 6.0 Å was employed.” (Page 2, Line 83).
- We revised “the doped graphene/Mg(001) interface model were established.” as “the doped graphene/Mg(001) interface models were established.” (Page 11, Line 228-229)
- We revised “Here, Egraphene, EMg(001), Einterface represents the energy of graphene, 4-layer Mg(001) slab model and interface model respectively, and A represents the interface area.” as “Here, Egraphene, EMg(001) and Einterface represent the energy of graphene, 4-layer Mg(001) slab model, and interface model, respectively, and A represents the interface area.” (Page 13, Line 253-254).
- We revised “In general, the Mg-1 layer gets the charge, and the Mg-2 layer, C1, C2 and C3 lose the charge.” as “In general, the Mg-1 layer gets the charge, and the Mg-2 layer, C1, C2, and C3 lose the charge.” (Page 13, Line 279-280).

Reviewer 2 Report
The revised manuscript incorporates the comments proposed. Thus the manuscript is recommended for publication.
However, a minor point needs to be made. The authors while referring to multiple papers tend to use [n,m] where n and m are reference numbers. It appears that they do not make a distinction between [n,m] and [n-m]. Do they mean only 2 references n and m or set of references between n and m.
They need to check on that. The same goes with [2,5] and also perhaps with Caragiu et al. [6, 9]. They may make a thorough check.
Author Response
Thank you very much for your valuable suggestions. According to your advice, we used [n-m] to represent a set of references from n to m.
- For example, Pooja Rani et al. [2-5] systematically studied the doping of B atom and N atom in graphene to adjust the band gap using DFT methods. (Page 1, Line 30-31)
- For example, Caragiu et al. [6-9] studied the interactions between alkali metals and graphene. (Page 1, Line 32-33)
- Also, transition and noble metals which were adsorbed on graphene sheets have been previously examined [10-15]. (Page 1, Line 33-35)
- It is believed that magnetic metals which are doped on graphene [16-21] have the abilities to induce large magnetic energy, or strong covalent bond formations, which can modify the electronic and magnetic properties of intrinsic graphene. (Page 1, Line 35-37)

This manuscript is a resubmission of an earlier submission. The following is a list of the peer review reports and author responses from that submission.
Round 1
Reviewer 1 Report
The paper on first principle study for interaction of Mg atom with metal atom doped or intrinsic graphene by Li et al presents a theoretical data by calculating the energy, electron density, electron density differences, and partial density of state of the composite systems. They do bring out some interesting results for the interactions of magnesium atom with Al-, Mn-, Zn-, and Zr-doped and intrinsic graphene, the doped metallic atoms which could greatly change the local structures of the graphene. It was observed that the Zn-doped graphene had the largest capability to capture the Mg atom, whereas the Mn-doped graphene displayed the lowest capability to capture the Mg atom. But they do not try to figure out why this difference between Mn doped graphene and Zn doped graphene takes place.
Further, I would have appreciated if the motivation of this study could be clearly mentioned.
They also observe the interactions of Mg(001) surface with Al-, Mn-, Zn-, and Zr-doped and intrinsic graphene (intrinsic and doped graphene/Mg interface), doped atoms can improve the interface adhesion work and interact with Mg layer to make graphene wrinkle, resulting in higher specific surface area of graphene and better stability of doped interface, perhaps leading to strengthening of composites of graphene. Could they add in their discussion some physical arguments in terms of why the behavior becomes different.
The authors may also note the following:
Your opening paragraph is short on literature survey for metal atom doping study as it does not include any references to B and N doping. ( the authors may survey for relevance some of the following: RSC advances 3 (3), 802-812, 2013, Physica E: Low-dimensional Systems and Nanostructures 62, 28-35, 2014, Advanced Science Letters 21 (9), 2826-2829, 2015, RSC advances 6 (15), 12158-12168, 2016.)
line 35 ref for aerospace and aeronautical applications missing.
I would be happy to consider the paper after suggested modifications.
Reviewer 2 Report
In this manuscript, the aim behind considering Mg atoms and surfaces for the study among several other species is not clear. The selection appears to be unreasonable, and the selection of doping atoms for graphene also emerges without a scientific consideration. Consequently, the manuscript appears to be incremental. All binding energies are computed with the PBE functional without considering comparisons with hybrid functionals.
The authors considered only one Mg atom above the intrinsic and doped graphene in order to investigate the influences of the doped graphenes on the Mg atoms. In this regard, what about the co-adsorption under the presence of several adsorbates?
There are many effects without explanation, such as:
Why did doping atoms increase the graphene's energy by approximately 2 to 16 %?
In table 1, the authors are comparing total energies, even when the number of electrons is different among systems. The authors must clearly explain why they are comparing the total energies of different systems.
Why were the doped atoms (Al, Mn, Zn, and Zr) greatly increased the 137 Eads?
Mulliken charges appear to give unreasonable results regarding relative electronegativities of dopant atoms. The authors must carefully revise their results.
Reviewer 3 Report
The manuscript seems to me rather interesting. It is devoted to the structural, energy, and electronic properties of systems consisting of magnesium atoms and doped graphene. Using a high-level DFT approach, the authors analyze their interface adhesion work, electron density, Mulliken populations, the density of states, etc. It cannot be said that the study is a breakthrough. However, the manuscript presents solid theoretical research. In my opinion, it does not contain serious drawbacks. However, authors should comment on the following issues before publication.
- First, I would like to hear more clearly from the authors about the presented study’s novelty. Many studies are devoted to graphene and its derivatives and their interaction with metal atoms. What is the contribution of the authors? So, the novelty should be more clearly highlighted in the Introduction.
- In addition, I have questions about the details of the calculations:
- The first task in any plane-wave calculation is the appropriate choice of the kinetic energy cutoff for wave functions. What value did the authors choose? Did the authors make the confirming convergence tests using increasing cutoff energy? This issue should be discussed in detail. What is the value of kinetic energy cutoff for charge density?
- For the accurate description of the processes studied, it is necessary to consider the van der Waals forces. Did the authors take into account, for example, Grimme’s D2 or D3 approach in their study or any other methods?
- What was the reason for the choice of the doped graphene supercells? Did the authors remake the calculations with larger supercells? Did the authors check the convergence of energy with changing supercell parameters?
- Did the authors check the chosen computational method? Did the authors check the chosen pseudopotentials, for example, on unsubstituted graphene?
3. Figure 8, in my opinion, is not very informative. Of particular interest is the density of states near the Fermi level. The authors, however, focused on deep levels. Such an arrangement of the figure does not allow us to understand, for example, whether pure graphene is a gapless semiconductor within the framework of the level of theory chosen by the authors.